# A CRISPR Path to Finding Vulnerabilities and Solving Drug Resistance: Targeting the Diverse Cancer Landscape and Its Ecosystem

*Benjamin McLean, Aji Istadi, Teleri Clack, Mezzalina Vankan, Daniel Schramek, G. Gregory Neely,\* and Marina Pajic\**

Cancer is the second leading cause of death globally, with therapeutic resistance being a major cause of treatment failure in the clinic. The dynamic signaling that occurs between tumor cells and the diverse cells of the surrounding tumor microenvironment actively promotes disease progression and therapeutic resistance. Improving the understanding of how tumors evolve following therapy and the molecular mechanisms underpinning de novo or acquired resistance is thus critical for the identification of new targets and for the subsequent development of more effective combination regimens. Simultaneously targeting multiple hallmark capabilities of cancer to circumvent adaptive or evasive resistance may lead to significantly improved treatment response in the clinic. Here, the latest applications of functional genomics tools, such as clustered regularly interspaced short palindromic repeats (CRISPR) editing, to characterize the dynamic cancer resistance mechanisms, from improving the understanding of resistance to classical chemotherapeutics, to deciphering unique mechanisms that regulate tumor responses to new targeted agents and immunotherapies, are discussed. Potential avenues of future research in combating therapeutic resistance, the contribution of tumor–stroma signaling in this setting, and how advanced functional genomics tools can help streamline the identification of key molecular determinants of drug response are explored.

## 1. Introduction

Cancer is among the most common lethal diseases in the Western societies.[1] Its development is broadly characterized through proto-oncogenes, genes which when amplified or activated through mutations, drive uncontrolled cell proliferation, cellular transformation and carcinogenesis, and tumor suppressor genes, which when genetically deleted or inactivated through mutation, further promote uncontrolled cell growth or genomic instability within developing tumors. Cancer, by virtue of being a disease caused by the accretion of mutations and genome alterations, is highly amenable to experimental approaches that involve genetic manipulation, and such studies can identify novel druggable targets, improve existing therapeutic response or overcome spontaneous or treatment-induced acquired resistance.

Earlier genome engineering efforts utilized sequence-specific gene

B. McLean, A. Istadi, M. Pajic
The Kinghorn Cancer Centre
The Garvan Institute of Medical Research
384 Victoria St, Darlinghurst, Sydney, New South Wales 2010, Australia
E-mail: m.pajic@garvan.org.au

T. Clack, M. Vankan, G. G. Neely
Dr. John and Anne Chong Lab for Functional Genomics
Charles Perkins Centre
Centenary Institute
University of Sydney
Camperdown, New South Wales 2006, Australia
E-mail: greg.neely@sydney.edu.au

D. Schramek
Centre for Molecular and Systems Biology
Lunenfeld-Tanenbaum Research Institute
Mount Sinai Hospital, Toronto, Ontario M5G 1X5, Canada

D. Schramek
Department of Molecular Genetics
Faculty of Medicine
University of Toronto
Toronto, Ontario M5S 1A8, Canada

M. Pajic
St Vincent's Clinical School
Faculty of Medicine
University of NSW Sydney
Sydney, New South Wales 2052, Australia

**Figure 1.** Basic outline of the CRISPR/Cas9 approach and in vitro screening workflow. A) CRISPR/Cas9 systems allow precise deletions or insertions at specific loci in the genome. Double strand breaks (DSBs) are generated under the guidance of single guide RNAs (gRNAs), which are then repaired by endogenous homology directed repair (HDR) or nonhomologous end-joining (NHEJ). In CRISPR interference (CRISPRi), and CRISPR activation (CRISPRa), catalytically inactive Cas9 fused to either transcriptional repressor or activator is guided by a gRNA to the transcriptional start site of an endogenous gene to repress or promote transcription, respectively. B) Cancer cells expressing recombinant Cas9 are transfected with a focused or whole-genome gRNA library. Sequencing and analysis of cancer cells which have been exposed to drug treatment in vitro identifies depleted or enriched sgRNAs, and from these genes that confer either sensitivity or resistance. Addition of single cell sequencing platforms to CRISPR screening enables analysis of gene function and gene-to-gene interactions upon genetic perturbations in single cells.

disruption through protein-DNA interactions (including zinc finger and transcription activator-like effector nucleases).[2,3] In contrast, the clustered regularly interspaced short palindromic repeats (CRISPR)/Cas system efficiently utilizes a complimentary RNA molecule to target Cas9 to virtually any gene[4] where it then alters the target locus in any number of ways, depending on the Cas9 system used.[5]

The biology of gene modification through CRISPR-Cas9 is well established. CRISPR-Cas9 nucleases are used in gene engineering as they efficiently cleave target DNA and introduce a double-stranded break (DSB) at a specific DNA sequence guided by a short, 20-base single-stranded guide RNA (gRNA) that is complementary to a target sequence of interest.[6,7] The generated DSB leads to activation of DNA repair mechanisms, primarily nonhomologous end-joining (NHEJ), which during repair results in small sequence insertions or deletions (indels) that can lead to frameshift and loss-of-function of the gene of interest (**Figure 1**A). Based on the barcoded nature of target-specific guides and availability of next generation sequencing, this technology is particularly useful for pooled screening to identify critical cancer vulnerabilities, and evaluate mechanisms of drug resistance or susceptibility.[8] In these experiments, cells are typically transduced with pooled (genome-wide or focused) lentiviral gRNA libraries to generate mutant cell populations.[8] Cells are then left to expand or exposed to a selection step (e.g., chemotherapy or targeted therapy), gDNAs are amplified from the transduced cell's genomic DNA, and guide abundance is quantified by next generation sequencing as proxy of a transduced cell's proliferation under a given treatment compared to an untreated condition. Guides found enriched may indicate that the gene targeted is required for a drug's cytotoxic activity (i.e., resistance mutations), whereas guides that are depleted identify essential genes (drop out), or genes that protect from drug activities (i.e., a chemogenetic interaction of a gene mutation that enhances the activity of a chemical compound; see Figure 1B).

Most commonly, CRISPR screening involves the use of active Cas9 to induce indels as a mechanism to genetically ablate gene activity, however more recently modified Cas9 variants have been developed and these technologies also have utility in pooled screening. For example, CRISPR interference platform

(CRISPRi)[9,10] utilizes a catalytically inactive Cas9 (dCas9) fused to a transcriptional repressor (KRAB,[11] Dnmt3A,[12] Dnmt3L,[13] CRISPRoff[14]). In this configuration, dCas9 sterically suppresses transcription when coupled with guides that target dCas9 to a gene promoter or exons, while the transcriptional repressors further reduce gene expression by epigenetically silencing the locus.[14] Alternatively, in CRISPR activation (CRISPRa), dCas9 is fused to transcriptional activators (CRISPRa,[15] VP64,[16] P65, HSF1[15–18]) to activate expression of specific target genes.[19] These modified CRISPR approaches have helped identify directional genetic interactions between cancer genes,[20] novel targetable genes that promote tumor growth and resistance in melanoma,[21] or help highlight transcriptional epistasis as a central mechanism of breast cancer progression in vivo.[22] These genome engineering technologies are continuously evolving, with next-generation editing systems, including base[23,24] and prime editing,[25] Cas12a (Cpf1), Cas13,[26] Cas14-based[27,28] platforms, rapidly enhancing our available toolbox.[25,29]

## 2. The CRISPR/Cas9 Tool Kit for the Identification of Novel Anti-Cancer Targets

The medicine of the future promises to treat diseases such as cancer by tailoring interventions to the individual rather than the one-fits-all approach of most current cancer treatments. This approach, commonly referred to as precision medicine, relies heavily on massive parallel DNA and RNA sequencing technologies,[30,31] coupled to therapeutic interventions that are effective in the context of disease-specific molecular fingerprint, yielding significant clinical benefit[32,33] for some of the most difficult-to-treat cancers.[34–36] CRISPR/Cas gene-editing tools play an important role in the discovery of novel drug targets and cancer dependencies, with direct implications for the design of optimal treatment strategies in precision medicine.

Today there are multiple comprehensive short hairpin RNA (shRNA, i.e., an artificial short RNA sequence which targets gene expression through RNA interference)[37–40] and CRISPR guide libraries that have been used to generate valuable cell vulnerability information for hundreds of cancer cell lines at the whole genome scale[41–45] (https://depmap.org/portal). In addition to Project DRIVE (conducted at Novartis),[42] Project Achilles (led by Broad Institute),[43] and AVANA,[43] the Sanger Institute-led Project SCORE successfully performed large-scale functional genomic screens for therapeutic target identification using 324 human cancer cell lines from 30 cancer types.[41] The same study further estimated that for every established gene drug target, there are additional seven new potential therapeutic targets, which lack biological and clinical validation.[41] Collectively, these comprehensive efforts represent incredible resources for the mining of promising new targets. For instance, Lord, et al.[44] applied new computational approaches to effectively integrate these independently generated datasets (AVANA, DEPMAP, SCORE, and DRIVE) and identify 220 robust genetic pan-cancer dependencies that may support prioritization of therapeutic targets in cancer.

CRISPR platforms enable unbiased target discovery at a considerably faster rate than earlier tools, with systematic functional validation of the identified hits in cancer models often being the challenge. Thus, CRISPR screening efforts have likely already

yielded an opulence of novel targets with potentially untapped clinical utility. Interesting highlights in this space include a functional association of Werner syndrome ATP-dependent helicase dependency and microsatellite instability (MSI),[41,46] loss of enzyme methylthioadenosine phosphorylase (MTAP) in cancer cells and enhanced sensitivity to the inhibition of epigenetic regulator protein arginine methyltransferase 5 (PRMT5)[47,48] and more recently, the therapeutic potential of stimulating NOTCH signaling in cancers driven by inactivation of this critical tumor suppressor.[49]

In the first instance, the complementary work of Behan, et al.[41] and Chan, et al.[46] published in the same issue of *Nature*, identified WRN, a RecQ DNA helicase, as a synthetic lethal target in cancers which harbor MSI. MSI is a pattern of hypermutation caused by inactivation of the DNA mismatch repair system (MMR), a critical regulator of DNA replication fidelity.[50] MSI occurs in a broad range of cancers[51] and is of clinical significance, as patients whose tumors harbor MSI and a higher mutational burden may present with higher immunogenicity, and improved sensitivity to immune checkpoint inhibitors.[52] Of note, inactivation of target WRN through complementary RNAi and sgRNA functional approaches effectively induced double-stranded DNA breaks and promoted apoptosis and cell cycle arrest in vitro in MSI cancer models,[41,46] and significantly inhibited colorectal tumor xenograft growth in vivo.[41] WRN is largely undruggable due to its canonical, evolutionarily conserved function in preventing premature ageing and chromosomal instability in normal cells. Yet, drugs that inhibit WRN activity could still be developed into a future tailored therapy for MSI tumors, particularly as part of combination therapy with immuno-oncology agents, with carefully optimized dose scheduling. This could be especially pertinent in immunotherapy resistant MSI cancer settings, where limited treatment options are available.

Data mining of existing CRISPR and deep RNAi screening efforts can drive the discovery of multiple distinct synthetic lethal anti-cancer targets. Analysis of Project Achilles datasets by Kryukov, et al.[48] alongside independent pooled shRNA screens of 390 cancer cell lines by Mavrakis, et al.,[47] revealed that MTAP loss, and concomitant accumulation of substrate S-methyl-5′-thioadenosine (MTA), is responsible for inducing sensitivity to PRMT5 inhibition. The functional synthetic lethality interaction between MTAP deficiency and PRMT5 and its potential in the treatment of cancer has been recently reviewed.[5] Both landmark studies by Mavrakis, et al.[47] and Kryukov, et al.[48] effectively demonstrated that in a broad range of MTAP-deficient cancer cells, levels of PRMT5 inhibitory cofactor molecule MTA were already significantly elevated, directly affecting endogenous PRMT5 activity. Thus, the same cells displayed further hypersensitivity to PRMT5 blockade, measured through dramatically decreased cancer cell proliferation, viability and tumor growth in vivo.[47,48] Pharmacological inhibition of PRMT5 has unfortunately not yielded the same promising results in MTAP-deficient settings,[47] likely due to the mechanism of action of the existing small molecule inhibitors, and specifically based on their effects on the PRMT5 activating cofactor S-adenosyl-l-methionine (SAM) and the inhibitory cofactor MTA.[5] To address the challenges for the clinical implementation of MTAP-deletion-targeted precision medicine strategies, research in this area has expanded toward identification of novel druggable sites

on PRMT5 to improve future therapeutic targeting,[53] preclinical and early clinical development of new generation PRMT5 inhibitors with promising safety and early signals of clinical activity[54–56] (NCT05094336, NCT03854227), alongside a parallel exploration of targeting the closely-linked methionine adenosyltransferase $2\alpha$,[57,58] also entering Phase I trials in malignancies with loss of MTAP (NCT04794699, NCT03435250). It will also be important to carefully consider the role of the intricate tumor microenvironment in regulating and potentially limiting the long-term therapeutic responsiveness to these proposed treatment strategies, as recent work by Barekatain, et al.[59] indicates.

Screens exploring the heterogeneity of cells within cancers, for example those focused on cancer stem cells (CSCs), are also identifying key pathways and anti-cancer targets that can help address the problem of tumor recurrence. CSC tumorigenic populations are often highly resistant to cytotoxic therapies or are sufficiently unaffected by targeted therapies, thereby leaving behind subpopulations that can stimulate tumor regrowth.[60] A key CSC factor that has been revealed through these efforts is the yes associated protein (YAP) oncoprotein. Wang, et al.[61] identified Kruppel-like factor 11 (KFL11) as a negative regulator of the YAP transcriptional co-activator in osteosarcoma CSC. Low KLF11 was found to be associated with poor prognosis and chemotherapy response in patients. However, when a KLF11 agonist, thiazolidinedione, was applied in combination with chemotherapy, chemotherapy response was restored. Similarly, Dai, et al.[62] performed a CRISPR/Cas9 screen in triple negative breast cancer, which revealed Hippo pathway members SAV1 and FRMD6 as negative regulators of YAP. Low SAV1 and FRMD6 were associated with high-grade tumor types and predicted poor survival outcomes. The same screen also identified previously unknown prooncogenic functions for mTOR components, RICTOR and WDR59, of the PI3K/AKT/mTOR (PAM) signaling pathway. Co-pharmacological inhibition of YAP and mTOR, with torin1 and verteporfin respectively, blocked tumor growth in patient derived xenografts.

Other key CSC specific components have also been comprehensively characterized through integrated genomic approaches. Using a combination of genome-wide CRISPR, RNA, and chromatin immunoprecipitation (ChIP)- sequencing analyses, Lytle, et al.[60] identified a new role for nuclear hormone receptor retinoic-acid-receptor-related orphan receptor gamma (ROR$\gamma$) in promoting pancreatic cancer progression. Genetic or pharmacological block of ROR$\gamma$ dramatically reduced pancreatic cancer growth in both patient-derived and genetically engineered in vivo settings. Its role in promoting the aggressive nature of pancreatic cancer was further characterized in retrospective clinical samples. For many cancers, especially difficult to treat cancers, CRISPR screens of different tumor cell types are thus providing important information that can strengthen the potency and long-term efficacy of therapies.

The development of in vivo CRISPR screening platforms is further aiding in the identification of cancer drivers, their interactions and role in the multistep process of tumor initiation and progression in an intact, whole-body system[62–70] (**Figure 2**). As such, these systems enable functional evaluation of the essential nature of genes in a more complete in vivo microenvironment, which is known to directly influence the tumor cell behavior. The Loganathan, et al.[49] study for example, developed

a reverse genetic CRISPR screen to functionally characterize the role of ≈500 long tail of rare but recurrent gene mutations during head and neck cancer initiation in cancer-susceptible genetically engineered mouse models. Of the 15 tumor-suppressor gene targets identified, major hits shown to cooperate with known head and neck cancer driver mutations (PIK3CA, HRAS, TP53, and HPV16-E6/7) were ADAM10, AJUBA and downstream target of NOTCH, RIPK4, followed by NOTCH2 and NOTCH3. Two of the targets, ADAM10 and AJUBA, with mutations and loss of heterozygosity occurring in 28% of human head and neck cancers, were shown to act as haploin-sufficient tumor suppressor genes by promoting NOTCH pathway activation.[49] Moreover, the majority of screening hits were NOTCH target genes with potent tumor suppressive capability. Collectively, these rare recurrent mutations were shown to converge onto a common pathway, the NOTCH signaling pathway and demonstrates its critical role in head and neck cancer pathogenesis, with potential future implications for patient treatment, another active area of further investigation.[71]

## 3. Functional Genetic Screens to Overcome and Circumvent Cancer Drug Resistance

In cancer, treatment response is limited by both intrinsic and acquired resistance. Understanding the biology behind treatment response and resistance may therefore lead to improved therapeutic strategies. Major research focus in this area is on the characterization of 1) essential mediators (and associated biomarkers) of drug resistance and 2) finding optimal ways to pharmacologically target these mechanisms using clinically useful agents.

An interesting highlight in this context is vemurafenib, the first clinically-approved targeted agent for the treatment of BRAF(V600E)-mutant advanced melanoma. Although vemurafenib presents one of the targeted break-through treatments in this setting, with demonstrated dramatic improvement in patient progression-free survival,[72–74] effects of this small molecule inhibitor were often found to be short-lived in metastatic melanoma, due to the rapid onset of resistance. The first defined multifactorial mechanisms of resistance to BRAF inhibition include RAS and receptor tyrosine kinase-mediated activation of cellular survival pathways,[75–77] with successful translational investigations of the subsequent proposed combination treatments, including dual BRAF–MEK-inhibitor regimens (reviewed in ref. [78]). In addition to these critical mechanisms, an unbiased genome-wide CRISPR knockout approach employed by Shalem, et al.[8] identified a range of new mediators of resistance to BRAF targeting in melanoma. Significant new hits from this seminal work included tumor suppressor neurofibromin 2 (NF2), Cullin 3 E3 ligase (CUL3), and members of the STAGA histone acetyltransferase complex (TADA1 and TADA2B). Key established markers of resistance against vemurafenib, NF1, and MED12[79,80] were also confirmed.[8] Several newly identified hits have been further validated, with a subsequent high resolution CRISPR mutagenesis screen of the noncoding regions surrounding NF1, NF2, and CUL3 genes providing new insight on the direct role of *CUL3* gene regulatory elements in promoting vemurafenib resistance.[81] Loss of CUL3 in vemurafenib-resistant melanoma was further found to associate with stabilization of the oncogenic SRC receptor tyrosine kinase, leading to activation

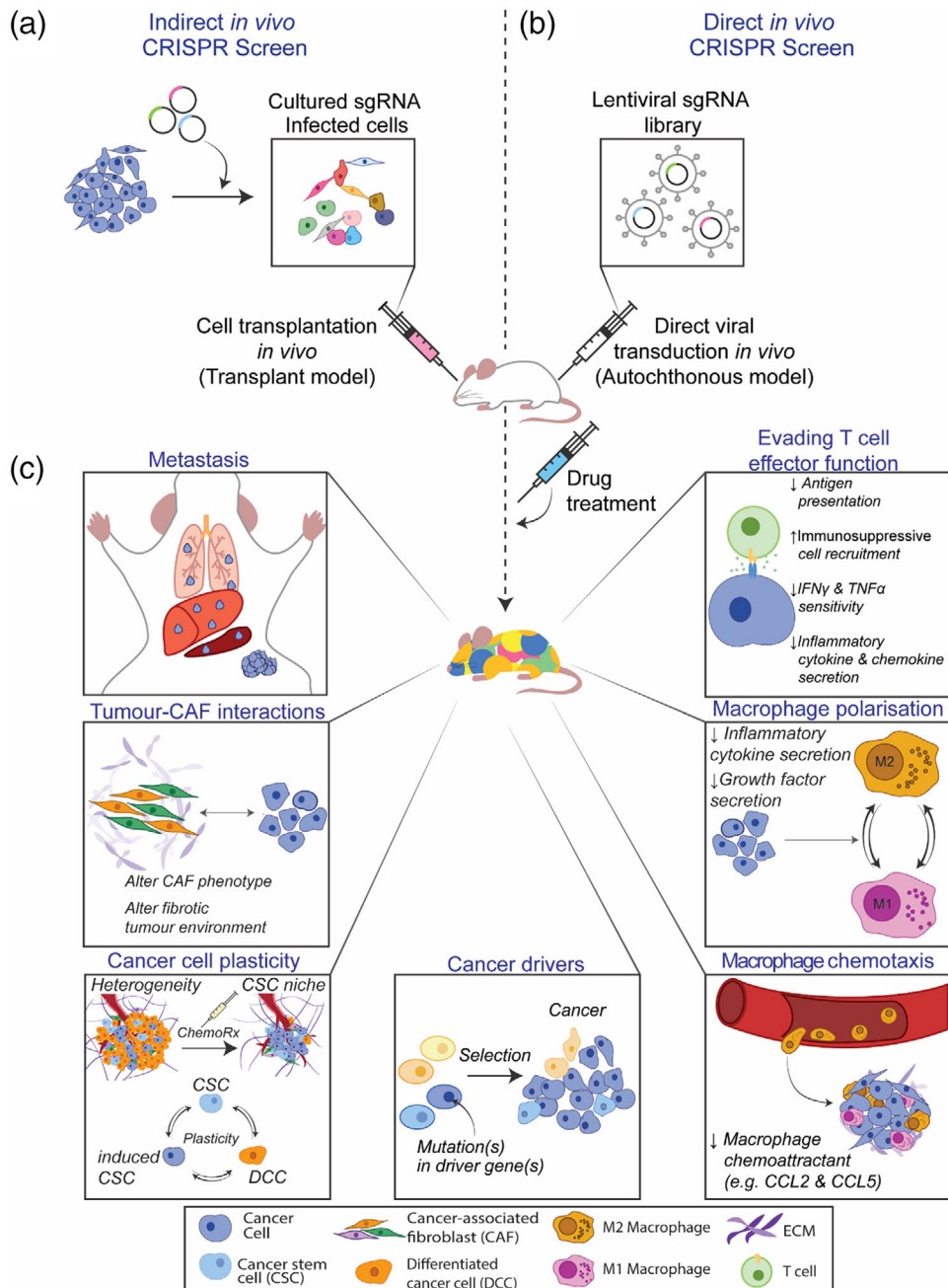

**Figure 2.** Schematic of tailored in vivo CRISPR screens for the identification of relevant cancer targets. A) For indirect in vivo CRISPR screens a tumor cell population is transformed with an sgRNA library in vitro prior to transplantation into mice. The subsequent enrichment of distinct mutants indicates potential candidate genes with functional significance in vivo. B) A lentiviral sgRNA library is injected directly into the target organ for direct in vivo CRISPR screens. C) Applications of in vivo CRISPR screens span diverse areas of cancer research, including the identification of cancer drivers, mediators of tumor evasion as well as anti-tumor immunity, broad therapeutic resistance, and mechanisms of metastasis.

of RAC1, and downstream MAPK signaling, effects that could be reversed with a SRC family kinase inhibitor, saracatinib.[82] Using the Brunello CRISPR sgRNA library, a similar screening platform for genetic dependencies in the presence of vemurafenib identified 19 previously reported genes and 14 additional novel hits,[83] converging on alterations in the MAPK pathway, epigenome and cell cycle, that collectively promote resistance to

vemurafenib in melanoma cells. Comprehensive characterization of these novel targets is a clear next step in understanding their clinical utility in frontline-treatment selection for patients with BRAF-mutant melanoma, as biomarkers of treatment resistance, and their diagnostic potential.

While CRISPR analyses for mechanisms of drug activity and resistance have primarily focused on screening using a single

drug, efforts to characterize the mechanisms of groups of anti-cancer drugs in the same genetic context have also been taken. For example, an in-depth characterization of mechanisms of action and resistance for 27 DNA-damaging agents was performed in retinal pigment epithelium-1 (RPE1) cells.[84] This massive study identified 890 genes that, when targeted, can alter cell survival in response to DNA damaging agents. There can be both direct and indirect mechanisms for why a given gene alters drug sensitivity, however one would anticipate many of these new regulators would participate in the DNA damage response in some capacity. Functional validation in this study confirmed a role for ERCC6L2 in the NHEJ pathway.[84]

While most pooled CRISPR screening efforts so far have focused on functional evaluation of coding genes, these technologies can also modulate expression of long noncoding RNAs (lncRNAs), which for the majority of transcripts have been notoriously difficult to establish any biological function.[85] To investigate the role lncRNAs play in chemotherapy resistance in acute myeloid leukemia, Bester, et al.[86] developed the dual protein-coding and noncoding Integrated CRISPRa screening (DICas) platform. Using a library of 88 444 sgRNAs targeting the transcriptional start site of 14 701 lncRNA genes, and selecting with the chemotherapy drug cytarabine, they identified lncRNAs that led to hyperactivation of the GAS6/TAM pathway, a resistance mechanism in many cancers, highlighting the potential importance of lncRNAs in understanding cancer and drug resistance.[86]

One direction of intense research is screening for chemogenetic interaction, where gene:drug interactions synergize to kill or suppress killing target cells are identified in order to understand pathways that can in combination target cancer. For example, exploiting chemogenetic interaction between mutations in the high-fidelity homologous recombination (HR) DNA repair pathway and increased cellular sensitivity to inhibitors of poly-ADP-ribose polymerase (PARP-i) has made remarkable impact on the clinical management of a range of solid cancers.[87–90] As such, the PARP - HR axis presents another classical illustration of an actionable biologic subtype, and clear example of the promise of the precision medicine paradigm.[87] Despite the obvious success, long-term PARP-i efficacy, even in the context of HR deficiency, appears to be limited by both intrinsic and acquired resistance. Understanding the genetic mechanisms underlying this resistance, and simultaneous identification of vulnerabilities that may reverse or circumvent resistance, has become a major research focus and the availability of pooled CRISPR screening has empowered these activities.

For instance, using genome-wide and high-density CRISPR-Cas9 screens, Pettitt, et al.[91] revealed novel mutations within and around the DNA-binding regions of PARP1 that can effectively drive PARP-i resistance, by altering PARP1 trapping on the DNA and thus its activity. The mechanisms behind this form of resistance are distinct from other established causes (BRCA1 reversion, 53BP1 mutation, depletion of RADX),[87,92] as cells retain sensitivity to other DNA poisons, including platinum therapy and topoisomerase I inhibitors.[91]

Moreover, using complementary genome-wide knockout and activation screens, Clements, et al.,[93] revealed additional novel genetic determinants of PARP-i resistance, namely loss of ubiquitin ligase HUWE1, or the histone acetyltransferase KAT5, this

time in the context of *BRCA2*-deficiency. These hits, with distinct roles in the regulation of end resection at DSBs and subsequent repair,[93] are joined by the TP53 induced glycolysis and apoptosis regulator (TIGAR)[94] and Cyclin C,[95] as further putative biomarkers and modifiers of treatment response to PARP-i, as a result of other recent CRISPR screening initiatives. Collectively, diverse alterations that tumors acquire while on PARP-i therapy may thus play important roles not only in signaling the emergence of treatment resistance but will likely also have important implications for the selection of subsequent rounds of therapy.

Application of CRISPR/Cas9 gene editing platforms have played an important role in the identification of new resistance mechanisms to conventional treatments including endocrine therapy and mainstay chemotherapy. One way in which secondary resistance can appear during treatment is through epigenetic alterations that result in significant chromatin reprogramming,[96] altered cell lineage fidelity,[97] and activation of alternative signaling pathways.[98,99] In estrogen receptor (ER) positive breast cancer, acquired resistance to endocrine therapy is associated with cancer spread, frequently following a period of tumor dormancy, with the appearance of metastasis eventually leading to death. Significant work has thus been undertaken to understand the mechanisms that drive endocrine therapy resistance, with CRISPR genome editing platforms enabling an unbiased approach to help solve this complex clinical problem. Genome-wide knockout screening efforts have revealed "bypassing" of negative feedback loops driven by C-terminal SRC kinase (CSK) as a critical mechanism of acquired cellular resistance to selective ER modulators and degraders, tamoxifen, and fulvestrant.[100] Using secondary CRISPR screens, the same study identified novel synthetic lethal vulnerabilities to overcome such resistance, by targeting p21 protein-activated kinase 2 (PAK2), downstream of CSK, in combination with an ER antagonist. Several preclinical studies have demonstrated significant potential of small molecule inhibitors of PAK2 as therapeutic agents, with development of selective PAK2 inhibitors underway, thus far only compounds targeting other PAK family members entering clinical trials (NCT00616967, NCT04281420, NCT00932126). Thus, full clinical implications of the proposed combination therapy approach to target endocrine resistance[100] remain to be investigated.

More recently, global alterations in the chromatin landscape of ER-positive breast cancer have been characterized as a major driver of clinical resistance to ER-targeting therapy. an Using an epigenome CRISPR knockout screen, seminal work by Xu, et al.[101] has identified loss of ARID1A, a member of the SWI/SNF chromatin remodeling family, as a major driver of clinical resistance to the ER antagonist and selective ER degrader fulvestrant in breast tumors.[101] Mechanistically, loss of ARID1A, was shown to cause major epigenetic remodeling of transcriptional networks during breast cancer progression, leading to a fundamental switch in cell state, from ER-dependent (and fulvestrant-responsive) luminal cells to an ER-independent basal-like breast cancer cell type. The same lineage switch was found to occur in breast tumors with ARID1A loss, and ARID1A genetic alterations in patients were associated with clinical resistance to selective ER degraders,[101] highlighting the direct clinical relevance of these findings. Importantly, this major study highlights the role of cellular phenotype switching as a

relevant modifier of therapeutic response, with the resulting transition from a drug-susceptible to drug-refractory tumor phenotype having significant implications on future combination therapy design. Although not considered directly druggable, ARID1A deficiency in cancer has been associated with perturbed regulation of the antioxidant system, namely the balance of intracellular glutathione and reactive oxygen species levels, thus rendering ARID1A-deficient cancer cells vulnerable to further inhibition of the glutathione metabolic pathway.[102] With small molecule inhibitors (e.g., APR-246) in clinical trials, further optimization of glutathione-targeted therapeutic interventions in ARID1A-deficient cancer settings is warranted.

Functional genomic screens are also proving useful tools in the identification of the drivers of cancer relapse as well as broad markers of cancer multi-drug resistance (MDR). For instance, we performed comprehensive genetic mapping of drug resistance to a spectrum of 27 anti-cancer treatments, ranging from clinically used chemotherapies to targeted agents in development.[103] As a powerful tool to study genotype-to-phenotype associations, these whole genome resistance screens confirmed several established drug resistance mechanisms (for example, SLC19A1, associated with methotrexate resistance, deoxycytidine kinase and resistance to pyrimidine nucleotide analogues), as well as uncovering new ones, for example, histone demethylase KDM1A and large neutral amino acids transporter SLC43A2 as mediators of vinorelbine and oxaliplatin sensitivity, respectively.[103] Moreover, by overlapping these data sets, we could identify common drug resistance pathways, and beyond well characterized cell death genes like PMAIP1/NOXA[104] and BAX,[105] we also identified the previously uncharacterized gene *c1orf115* which we named "required for drug-induced death 1" or RDD1. RDD1 is a key determinant of broad resistance to anti-microtubule agents (vincristine, vinorelbine, docetaxel), DNA synthesis inhibitor cladribine, and tyrosine kinase inhibitor imatinib. Importantly, RDD1 was found to be of broad prognostic relevance, with low RDD1 mRNA levels correlating with poor survival across a range of solid tumors, including ovarian, lung, gastric, liver, kidney cancers and sarcomas, as well as a marker of clinical response to paclitaxel.[103] The precise functional roles of RDD1 and other newly identified targets in cancer cell survival and resistance are yet to be elucidated, with detailed characterization underway. Moreover, examination of the effect of gene inactivation outside of a controlled genetic background (such as the HAP1 model system, or an isolated cancer cell line model), and across diverse cancer mutational landscapes, in models of increasing complexity (3D, in vivo), will enable investigation of complex genetic effects and interactions that may more accurately reflect the clinical picture of treatment failure.

## 4. The Complexity of Tumor Microenvironment Interactions in Driving Aggressive Phenotypes: CRISPR-Guided Characterization of Immunotherapy Resistance

As cancer develops, it creates its own ecosystem, a dynamic cancer microenvironment, which actively drives the process of cancer evolution toward more aggressive, invasive and treatment-resistant phenotypes.[106–108] The cancer microenvironment of diverse solid cancers consists of major elements: an extracellular matrix and diverse "corrupt" normal cells including infiltrating immune cells, cancer-associated fibroblasts, endothelial cells, pericytes, vascular and lymphatic channels.[109] In addition to targeting specific tumor cell aberrations they are designed against, selective small molecule inhibitors can have profound effects on diverse stromal elements within developing tumors, for example, by modifying the surrounding fibrotic matrix and improving drug penetrance into the tissue,[110–112] as well as in altering the immune-suppressive elements which drive cancer immune evasion to improve overall immunotherapy response.[113,114] Understanding the crosstalk between tumor cells and their unique environments is thus critical to developing truly effective combination therapies to target cancer.

A relatively recent application of CRISPR pooled screening technology in the treatment of cancer was the development of multiplexed activation of endogenous genes as an immunotherapy (MAEGI) by Wang et al. in 2019.[115] This new immunotherapy elicited antitumor immunity via multiplexed activation of endogenous genes within the tumors, directly changing the expression of endogenous genes and altering the tumor microenvironment. Established solid tumors of various cell types were injected with Adeno-associated virus containing dCas9 and a whole genome activation library, causing aberrant expression of immunogenic epitopes and leading to dramatic tumor shrinkage.[115]

Although systematic investigation of the genetic interactions that influence the cancer cell's ability to effectively evade the host immune system has been challenging, CRISPR genetic screening is emerging as a powerful platform in this space, particularly in the identification of new immuno-oncology targets (using a range of co-culture and in vivo screening platforms; Figure 2). A major highlight includes the work of Pan, et al.[116] whereby following genome-wide genetic perturbation of immunotherapy resistant B16F10 melanoma cancer cells and co-culture with syngeneic T cells, the authors identified new tumor cell-intrinsic mechanisms that regulate resistance to T cell-mediated tumor kill. In addition to previously characterized targets, including MHC class I, tumor necrosis factor and interferon-γ (IFN-γ) network components,[117,118] PTPN2, ADAR1, NFκB pathway genes[65,119] and APLNR mutations,[120] key newly identified hits included all three unique components of the SWI/SNF chromatin remodeling complex (Arid2, Pbrm1, and Brd7), which were shown to actively drive resistance to T cell-mediated killing. Decreased levels of ARID2 and PBRM1 correlated with elevated cytotoxic activity contributed by CD8 positive T cells in human cancers, and improved response to immune checkpoint blockade in the Pbrm1-deficient B16F10 animal model of melanoma.[116] Complementary work by Miao, et al.[121] demonstrated significantly improved clinical response to immune checkpoint inhibitors in patients with metastatic clear cell renal cell carcinomas that harbor loss-of-function mutations in PBRM1. Despite its established role as a tumor suppressor in renal carcinomas,[122] these studies highlight an additional, distinct role for PBRM1 and other components of the SWI/SNF complex in modulating cancer cell susceptibility to immune destruction, features which may have important implications for the clinical application of immuno-oncology agents.[121]

New gene networks that enable cancer cells to effectively avoid destruction by the immune system are continuously being discovered. For example, using a range of murine cancer models, in vitro co-culture, and in vivo CRISPR screens, Lawson, et al.[123] recently uncovered a conserved role for the autophagy pathway (top hits including family members Atg12/3/9a/7/12) in mediating cancer-intrinsic immune evasion within the developing tumor microenvironment. Pharmacological inhibition of autophagy in vitro using the VPS34 inhibitor autophinib led to increased T-cell induced kill across melanoma and colon cancer models,[123] suggesting a broader tumor-intrinsic role for this process in the regulation of tumor inflammation and immunotherapy resistance.

Additionally, other focused in vivo CRISPR screens have shown that conserved H3–H4 histone chaperone Asf1a actively promotes immunosuppression in a genetically engineered $Kras^{G12D}/Trp53^{-/-}$ mouse model of lung adenocarcinoma, with Asf1a deficiency significantly stimulating the effect of anti-PD-1 immune checkpoint blockade in murine models of lung and colon cancer.[68] Assessment of the diverse immune microenvironment of in vivo lung adenocarcinomas showed increased intra-tumoral accumulation of pro-inflammatory monocytes and macrophages, accompanied by enhanced M1-like macrophage polarization and T-cell activation as key mechanisms behind the observed efficacy.[68] In parallel, interrogation of genetic dependencies of in vivo mouse models of triple negative breast cancer has established chromatin regulator Cop1, as a critical new mediator of macrophage chemotaxis and infiltration in the tumor microenvironment, broad antitumor immunity, and response to immuno-oncology agents.[64] Our recent work has shown that SerpinB9 or the cancer-testis antigen Adam2 can dramatically alter the response of a KRas-mutant driven lung adenocarcinoma mouse model to adoptively transferred cytotoxic T-cells.[124] SerpinB9 was also recently identified as a general regulator of tumor immune evasion and potent cancer immunotherapy target.[125,126]

Co-culture models have become highly relevant tools as part of genome editing screens for the discovery of the intricate interactions between cancer cells and cytotoxic T cells.[15] Parallel use of co-culture and in vivo genetic screening, plus the introduction of 3D organoid platforms in this setting,[127] are poised to further increase the physiological relevance of the identified targets that mediate tumor immune evasion and drug resistance. Previously, genome-scale screens in organoids were hampered by technical limitations, including the infeasibility of obtaining the cell numbers required for whole-genome screens.[128,129] Recent improvements to organoid culture methods enable practical and scalable CRISPR screening of organoid cultures,[130] significantly reducing cost and technical demand. Additionally, Ringel et al.[131] described a method for sequencing individual sgRNA-transduced organoids that allows whole-genome screens to be performed without the need for large cell numbers.

These technological advances will enable more widespread utilization of CRISPR technologies in physiologically relevant 3D systems. Screening in human intestinal organoids showed that loss of ARID1A and other SWI/SNF chromatin remodeling components alters gene transcription downstream of TGF-$\beta$ signaling, preventing TGF-$\beta$-mediated growth regulation.[131] Additionally, a screen to identify resistance mechanisms to CDK4/6 inhibition revealed distinctly different gene hits when conducted in a 3D compared to a 2D setting. Specifically, $Slc39a6$ and $fam134b$ were significantly enriched in 3D, but not in 2D using the same breast cancer line. $Slc39a6$ was further validated to play a role in resistance to both CDK4/6 inhibition and fulvestrant, demonstrating the utility of a more physiologically relevant culture system in identifying novel resistance mechanisms.[132] Although not all of these highlighted genetic vulnerabilities are directly therapeutically targetable yet, CRISPR screens can effectively contribute to our understanding of the epigenetic drivers of tumor immunosuppression, broad therapeutic resistance and importantly, aid identification of novel predictive biomarkers of treatment response as well as new therapeutic strategies in the near future.

## 5. Conclusion

Both inter- and intra-tumoral heterogeneity still present an important complicating factor in the interpretation of findings from CRISPR screens. First, the genomic make-up of cancer is complex and continuously evolves from the early transformation events, stages of growth, relapse on-treatment, metastasis, and right through to end-stage disease. In turn, the complex mutational and epigenetic alterations and their dynamic interplay can have profound effects on the phenotype of the diverse cancer cells within the growing tumor, and importantly, the different cell types within its surrounding environment. Due to these mechanistic complexities of cancer, identification of genes, and networks that drive treatment resistance, and from these, clinical predictive biomarkers, has been notoriously difficult. Forward genetic screens have emerged as powerful tools that have dramatically improved our insight on the biology of cancer cells, anti-cancer drug resistance and tumor immune escape mechanisms, as well as in the identification of novel tumor vulnerabilities (Figure 2). As CRISPR methodologies and model systems continue to improve (3D tumoroids and more complex co-culture platforms, in vivo), there is potential to adapt these tools beyond the study of tumor cell—immune cell genotype—phenotype functional assessments and include other cell types, including diverse cancer-associated fibroblasts, perivascular, endothelial cells, and other essential elements of the intricate cancer microenvironment that promote cancer progression and therapeutic resistance.

With the close integration of single cell sequencing and complementary new technologies within CRISPR screening pipelines,[133,134] one can envisage that soon, systematic assessment of how complex mutational landscapes and tumor heterogeneity mediate intrinsic and acquired mechanisms behind clinical treatment failure will be entirely possible.

## Supporting Information

Supporting Information is available from the Wiley Online Library or from the author.

## Acknowledgements

M.P. received fellowship support from the Snow Medical Foundation and the Philip Hemstritch Foundation. M.P. and G.G.N. received funding from

the National Health and Medical Research Council of Australia (1162556, 2003310, 1162860, 1185002, 1158165, 1158164) and Cancer Institute New South Wales (1120910). M.P. is further supported by the Girgensohn Foundation. B.M. is recipient of the Australian Government Research Training Program (RTP) Scholarship. After initial online publication, the name of T.C. was corrected on December 12, 2022, as in the original version the name "Teleri" was incorrectly written as "Telery".

## Conflict of Interest

The authors declare no conflict of interest.

## Author Contributions

B.M. and A.I. contributed equally to this work. M.P., G.G.N., B.M., and A.I. wrote the article. B.M., A.I., T.C., M.V., and D.S. researched data and contributed to writing the article. A.I. designed the figures. All authors reviewed the manuscript, actively contributed to the discussion and editing of the manuscript prior to the submission.

## Peer Review

The peer review history for this article is available in the Supporting Information for this article.

## Keywords

cancer drug resistance, chemotherapy, CRISPR/Cas9, immunotherapy, predictive biomarkers

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

[48] G. V. Kryukov, F. H. Wilson, J. R. Ruth, J. Paulk, A. Tsherniak, S. E. Marlow, F. Vazquez, B. A. Weir, M. E. Fitzgerald, M. Tanaka, C. M. Bielski, J. M. Scott, C. Dennis, G. S. Cowley, J. S. Boehm, D. E. Root, T. R. Golub, C. B. Clish, J. E. Bradner, W. C. Hahn, L. A. Garraway, *Science* **2016**, *351*, 1214.

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

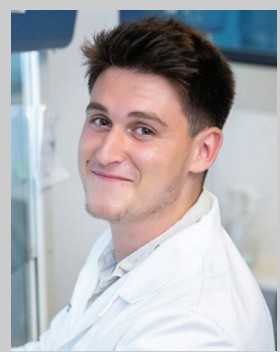

**Benjamin McLean** is currently completing his Ph.D. at the University of New South Wales, under the supervision of Dr. Marina Pajic and Dr. Sean Porazinski. His thesis investigates novel therapeutic targets in pancreatic cancer, looking in particular at how targeting the complex tumor microenvironment can benefit treatment resistance and alter disease progression. His aim is for this translational research to inform the development of novel combination therapies to improve patient responses in the clinic.

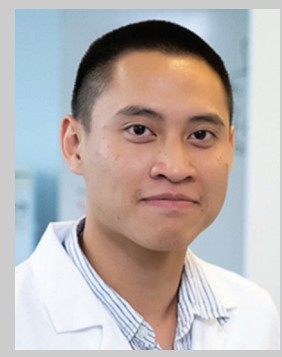

**Aji Istadi** completed his Master of Biomedical Science degree at the University of Melbourne under the supervision of Dr. Ilia Voskoboinik and Prof. Joe Trapani at the Peter MacCallum Cancer Centre. His research sought to improve the cytotoxic activity of human CAR T cells in chronic lymphocytic leukemia patients by modulating cytotoxic protein expression in CAR T cells. In his current work at the Garvan Institute he seeks to understand the biology underlying treatment responses in Pancreatic cancer, and develop novel, personalized immunotherapy approaches in this difficult-to-treat cancer.

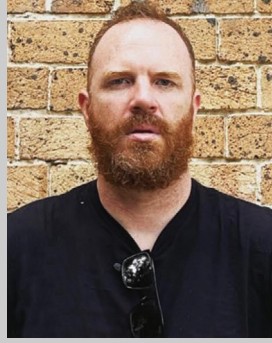

**Greg Neely** completed his Ph.D. in human immunology (AIDS related pathogens) at the University of Calgary, Canada (2004) and went on to train in functional genomics of pain at IMBA in Vienna, Austria (2003–2010). Since 2011 he has been running a lab in Sydney, Australia, and he is currently based at the Charles Perkins Center at the University of Sydney. His lab uses pooled CRISPR screening to investigate human cell biology with a focus on cellular mechanisms controlling pain.

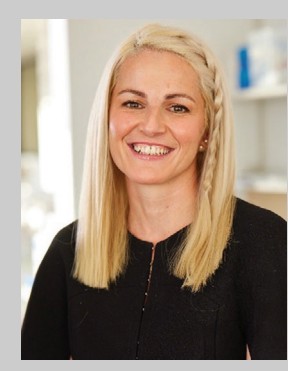

**Marina Pajic** completed her Ph.D. in cancer drug resistance at the University of NSW Sydney, Australia (2008), with further training at the NKI, Amsterdam (2007–2010). Since 2013, she has been running an independent lab at the Garvan Institute of Medical Research (Sydney, Australia), where she is the co-head of the Precision Cancer Medicine Program. Her lab studies the deregulation of molecules commonly hijacked in cancer which drive tumor i) genome instability, ii) invasion leading to metastasis, or iii) resistance to therapy. These insights are incorporated to develop novel precision medicine guided treatment strategies for cancer.

**2200014 (14 of 14)**

