## [**Supplementary Information**: Record of Transparent Peer Review · Advanced Genetics]

A CRISPR path to finding vulnerabilities and solving drug resistance: targeting the diverse cancer landscape and its ecosystem

Benjamin McLean, Aji Istadi, Telery Clack, Mezzalina Vankan, Daniel Schramek, Gregory Neely* and Marina Pajic*

* Corresponding authors

Date submitted: 01 Apr 2022

Editors: Kerstin Brachhold, Andrew L. Hufton

1 st Peer Review Decision	22 Jun 2022
-------------

Dear Dr Pajic,

Thank you for submitting your manuscript entitled "A CRISPR path to solving drug resistance: targeting cancer drivers and their ecosystem" (Review, No. ggn2.202200014) to Advanced Genetics. The reviewer report and comments are included at the end of this e-mail.

On the basis of these reviewer comments, we are not able at this stage to accept your manuscript for publication. I invite you to address the reviewer comments and make the necessary changes and improvements in a major revision of your manuscript.

My apologies that the review process has taken so long. Unfortunately, I had a difficult time in finding reviewers. I had to send close to 30 invitations to finally obtain these two reports below. As you can see, the report from reviewer 1 is rather short. That reviewer only mentions that some of your claims require better referencing. This point is also picked up by reviewer 2. This second reviewer gives a much more detailed report and identifies some major points that you need to carefully address.

Further, it is the author's responsibility to request permission to reproduce or adapt previously published figures. The copyright of figures generally belongs to the publisher of the journal where they first appeared. This also applies to your own previous publications. If you have not obtained permission to reproduce the figures yet, please do so before you submit your revised manuscript. We can only proceed with the publication once you have received all necessary permissions.

To submit your revision, go to <https://www.editorialmanager.com/advgenet/> and log in as an Author

using your username (*****) and password. Your submission can be found under the menu item "Submissions Needing Revision". The changes to your manuscript should be highlighted in a different color in the primary "Revised Manuscript" file.

Please provide a point-by-point response letter addressed to the reviewers, including a list of changes made and a rebuttal to any comments with which you disagree. You may copy the letter into the "Respond to Reviewers" box (if it is plain text only) or upload it as a "Response Letter to Reviewers" item (if it contains figures, tables, or special formatting such as formulas or equations). If necessary, you may also upload a separate revision cover letter addressed to the editor with any other information not intended for reviewers as a "Cover Letter to Editor" item. You will also be asked to upload a .zip archive containing the production data that will be used if your manuscript is accepted. See below for more details.

We should receive your revised manuscript by 29 Jul 2022. When we receive your revised manuscript, its suitability for publication in *Advanced Genetics* will be reassessed.

We recognize that authors are doing their best to revise manuscripts under challenging circumstances due to the COVID-19 pandemic. Should you need extra time, do not hesitate to contact the editorial office.

Yours sincerely,

Kerstin Brachhold

P.S. Please help avoid delays by referring to the Manuscript Preparation Checklist (<http://www.advgenet.com/authorguidelines>) and use the appropriate article template when preparing your revised manuscript. Please also follow the instructions to prepare and upload your Production Data materials. These include: the full, non-highlighted text of your manuscript (with all figures with a resolution of at least 300 dpi, schemes, and tables) in editable format - Word DOC/X or LaTeX; a short summary (50-60 words) and an eye-catching color image for the Journal's Table of Contents; and, if applicable, Supporting Information*.

*The Supporting Information document(s) will be published alongside your article and should be non-highlighted and ready for publication. Video formats may also be included.

Copyright information should be included in each figure caption as follows (the reference number [REF] should be superscripted):

Reproduced (Adapted) with permission.[REF] Copyright YEAR, Publisher Name.

Or, for figures reproduced or adapted from CC-BY open access publications (insert the name of the specific CC-BY license for XXX; permission should be obtained from the Copyright Holder for CC-BY-NC licenses):

Reproduced (Adapted) under the terms of the XXX license.[REF] Copyright YEAR, Copyright Holder Name(s).

REVIEWER REPORT:

Please note that reviewers may not be numbered consecutively. Where reviewers have provided additional files, these are available here: *****

EVALUATION:

Reviewer's Responses to Questions

Is the topic timely?

Reviewer #1: Yes

Reviewer #2: Yes

Does the manuscript contain a critical but fair evaluation of the literature?

Reviewer #1: Yes

Reviewer #2: Partially

Does the manuscript provide a new and insightful perspective?

Reviewer #1: Mostly

Reviewer #2: Partially

Is the manuscript balanced, accurate and complete?

Reviewer #1: Yes

Reviewer #2: Partially

Which aspects of scholarly presentation require improvement (if any)?

Reviewer #1:

*References

Reviewer #2:

*Clarity

*Manuscript structure

*Display items

*References

COMMENTS TO AUTHOR:

Reviewer #1: This is well written and informative review concerning to application of CRISPR gene editing technology in drug discovery for cancer oncogenes. Totally speaking, it is worthy to be published!

Just a minor comment:

In the introduction section, authors need to provide more reference information. As in several parts, there are no reference provided.

Reviewer #2: In manuscript "A CIRPSR path to solving drug resistance: targeting cancer drivers and their ecosystem", McLean at al. give a timely and important overview of two overlapping, but not identical fields of research - leveraging large scale CRISPR screens to dissect cancer vulnerabilities on one hand, and resistance markers, on the other. The authors start by introducing cancer as a fundamentally genetic disease, amenable to genetic interrogation, and provide a short overview of CRISPR biology and its use in screening strategies to identify resistance-mediating genes. The main text is then divided in three sections, the first of which discusses identification and dissection of cancer vulnerabilities through viability CRISPR screens. The other two sections are dedicated to the literature about uncovering resistance mediators, either to classical and targeted therapies (curiously called "multi-drug resistance") or to immunotherapies.

The main tension in the text is already evident in the title - the authors suggest that the path out of cycles of resistance is to target cancer drivers. Astonishingly, this claim is never explored in the text itself. It is not self-evident at all that targeting cancer drivers will lead to less resistance, indeed resistant phenotype is result of many therapies that target drivers, as is discussed throughout the whole manuscript. In other words, it is largely unclear to this reviewer what connects section 1 to sections 2 and 3 of the manuscript, as finding new vulnerabilities often has nothing to do with finding causative agents of resistance. The whole logic of adaptive therapy, for example, is that selection pressures of

growth (drivers) and resistance are different enough to make therapy aimed at balancing these requirements possible. This is a serious conceptual problem for the text, as the segments seem additive, but in a random fashion.

The second major issue I see with the text is that it is often under-referenced. Many claims contain no references, and some only partial. I am listing here some representative cases: page 5 lines 47-48, CRISPRi, and page 6 line 1, CRISPRa give one reference for each, though there are multiple systems in use; page 6 line 13-14, no references for base editing (only prime), Cas13 nor Cas14 platforms; page 6 line 50, no reference for shRNA libraries, what is more, there is even no explanation what shRNAs are; page 10 line 55 introduces in vivo screens without any references.

The third major issue is the nomenclature of the screens that seems confusing: starting from Figure 1b, the text labels screens "positive" and "negative". This is misleading, as it might seem to mean "positive selection" and "negative selection" screens, but this is not what the authors mean. They mean enriched and depleted hits, which has nothing to do with the way how screens were performed. Thus, page 10 lines 8-25, discuss an in vivo viability CRISPR screen and mention hits from "positive screen" and "negative screen". Of course, only one screen was actually performed, and what is discussed are positively and negatively selected genes, not different screens. Since both positive selection CRISPR screens (for example drug screens) and negative selection CRISPR screens (for example viability screens) can have both positively and negatively selected genes, nomenclature offered in the text should be corrected.

Minor issues:

- 1.) Page 5, line 47, KRAB domain should be capitalized, mouse domains Dnmt3a and Dnmt3L listed both for CRISPRoff, not just Dnmt3a.
- 2.) While in vivo screens discussed in sections 2 and 3 offer good examples, the progress made in organoids is entirely lacking. Co-culture models are only mentioned in passing in concluding remarks, but I think they should be discussed in more detail. In fact, screening in organoids has now been done for different purposes, including finding cancer vulnerabilities, and could also be discussed in the text.
- 3.) Page 6 line 1, components of CRISPRa are said to "epigenetically activate" expression of target genes. At least in case of p65 and VP64 this is not true to my knowledge, as they contact basal transcriptional machinery and do not work through epigenetic activation. If other evidence exists for this claim, it should be properly cited.
- 4.) Page 20, line 47-48, mentions co-culture and 3D organoids, but the referenced paper was not performed in co-culture. Reformulate.
- 5.) The title of section 2 uses the term "multi-drug resistance", which I found confusing, as the real multidrug resistance was discussed in only few cases. The term means resistance to different drugs at the same time, a setting rarely established and almost never tested in literature under discussion.

--

Dr. Kerstin Brachhold, Deputy Editor
Advanced Genetics
E-mail: AdvGenet@wiley.com
Tel: +49(0)6201-606-362

<http://www.advgenet.com>

Authors' Response to 1st Review

12 Sept 2022

Dear Dr Kerstin Brachhold,

Re: Advanced Genetics - New Submission of Revised Manuscript 'A CRISPR path to solving drug resistance: targeting the diverse cancer landscape and its ecosystem' (Review, No. ggn2.202200014)

We would like to thank you for considering our manuscript in the initial round and to the Reviewers for their thorough examination of our work and constructive comments. We have taken into careful consideration all reviewers' comments and we believe our newly revised manuscript is now in an excellent position to be considered for publication in *Advanced Genetics*.

We summarised the comments/issues raised by the reviewers and how we addressed them:

Reviewer 1 comment: "In the introduction section, authors need to provide more reference information. As in several parts, there are no reference provided" We thank reviewer 1 for this comment. We have provided additional references in the introduction of the revised manuscript, specifically, references 2, 3, 6, 7, 9-13, 15-18, and 23-27 in this section.

-----Reviewer 2-----

Reviewer 2 major comment 1 was with regard to exploring the claim that targeting cancer drivers leads to reduced treatment resistance. "The main tension in the text is already evident in the title - the authors suggest that the path out of cycles of resistance is to target cancer drivers. Astonishingly, this claim is never explored in the text itself. It is not self-evident at all that targeting cancer drivers will lead to less resistance, indeed resistant phenotype is result of many therapies that target drivers, as is discussed throughout the whole manuscript." We, the authors, do not believe that we make the claim suggested by Reviewer 2. The point which we as authors are aiming to get across to the reader is simply that the new CRISPR screening technologies have uncovered new important roles for numerous genes that mediate complex cancer processes including tumour-microenvironment interactions, disease progression, metastasis and chemoresistance. The importance of these emerging targets is yet to be fully understood and should not be underestimated. Ultimately,

to define these novel targets as ‘cancer drivers’ or not is up to the discretion of the reader. **“In other words, it is largely unclear to this reviewer what connects section 1 to sections 2 and 3 of the manuscript, as finding new vulnerabilities often has nothing to do with finding causative agents of resistance”**. Again, the authors wish to express simply that CRISPR technologies can identify previously unknown mechanisms involved in cancer chemoresistance (‘vulnerabilities’) and indeed have been profitably utilised in multiple cited studies to identify such mechanisms, the inhibition of which may reverse chemoresistance. In section 1 (introduction) it is clearly stated that *‘Guides found enriched [in CRISPR screens] indicate that the gene is required for a drug’s cytotoxic activity (resistance mutations), whereas guides that are depleted identify essential genes (drop out), or genes that protect from drug activities’* (page 4, lines 85-87 of current manuscript). Additionally, whether a given gene is a ‘new vulnerability’ or a ‘resistance mechanism’ is not really relevant, as from a pragmatic point of view both ‘new vulnerabilities’ and ‘resistance mechanisms’ could be targeted to successfully improve disease outcome. **“The whole logic of adaptive therapy, for example, is that selection pressures of growth (drivers) and resistance are different enough to make therapy aimed at balancing these requirements possible. This is a serious conceptual problem for the text, as the segments seem additive, but in a random fashion”**. The object of the review was merely to provide an overview of recent developments in CRISPR technology, not to make broad hypotheses which question the logic of adaptive therapy, which Reviewer 2 suggests. The claims which Reviewer 2 has suggested we are making were never the goal of this review, thus for clarification purposes we have changed the ‘cancer drivers’ to ‘the diverse cancer landscape’ in the title of this review.

Reviewer 2 major comment 2 was with regard to under-referencing. Reviewer 2 lists the following representative cases of under-referencing in the text:

Page 5 line 47-48, regarding CRISPRi: Additional references have been supplied (references 9-13 of revised manuscript. See page 4 lines 93-94).

Page 6 line 1, regarding CRISPRa: Additional references have been supplied (references 15-18 of revised manuscript. See page 5 lines 98-99).

Page 6 line 13-14, regarding “no references for base editing (only prime), Cas13 nor Cas14 platforms”: Additional references have been supplied. Specifically, references 23 and 24 have been added for base editing; reference 25 has been added for Cas13 editing; and references 26 and 27 added for Cas14-based platforms. See page 4, line 104 of revised manuscript.

Page 6 line 50, regarding “no reference for shRNA libraries, what is more, there is even no explanation what shRNAs are”: An explanation of shRNA is given on page 5, lines 116-117 of the revised manuscript, with additional references (37-40).

Page 10 line 55, regarding in vivo screens: Additional references have been supplied (references 62-70. See page 9, line 208 of revised manuscript).

In addition to the references added above, references have also been added in other parts of the text not pointed out by Reviewer 2. Page 13, lines 313-314 of revised manuscript includes references 96-99 (concerning epigenetic mechanisms of treatment resistance). Page 16, lines

378-384 now includes additional references regarding the complex cancer microenvironment (references 106-109, 112).

Reviewer 2 major comment 3 was concerning the nomenclature of the screens. We agree that the nomenclature may cause confusion. Specifically, the terms ‘positive’ and ‘negative’ with regards to screens could mean ‘positive selection’ and ‘negative selection’ screens, whereas the intended meaning of ‘positive’ and ‘negative’ in this context is ‘enriched’ and ‘depleted’ genes. Therefore the text on page 8, lines 192-195 of the revised manuscript has been altered to remove the terms ‘positive’ and ‘negative’ as descriptors of CRISPR/Cas9 screens to avoid this potential ambiguity.

Reviewer 2 offered the following minor issues:

Page 5, line 47, KRAB domain should be capitalized, mouse domains Dnmt3a and Dnmt3L listed both for CRISPRoff, not just Dnmt3a. These corrections have been applied (see page 4, line 94 of revised manuscript).

While in vivo screens discussed in sections 2 and 3 offer good examples, the progress made in organoids is entirely lacking. Co-culture models are only mentioned in passing in concluding remarks, but I think they should be discussed in more detail. In fact, screening in organoids has now been done for different purposes, including finding cancer vulnerabilities, and could also be discussed in the text. The authors agree, and thank reviewer 2 for their suggestion. We have extended our comments to include remarks about recent innovations in organoid screens and notable findings revealed by this model (see page 19, lines 446-464 of revised manuscript. Additional references 128-132).

Page 6 line 1, components of CRISPRa are said to "epigenetically activate" expression of target genes. At least in case of p65 and VP64 this is not true to my knowledge, as they contact basal transcriptional machinery and do not work through epigenetic activation. If other evidence exists for this claim, it should be properly cited. The authors agree that the mechanism by which these components activate gene expression is not necessarily epigenetic. The text has been modified accordingly to remove term ‘epigenetic’ (page 5, line 99 of revised manuscript).

Page 20, line 47-48, mentions co-culture and 3D organoids, but the referenced paper was not performed in co-culture. Reformulate. An additional reference has been added which includes co-culture studies (Reference 15, page 19, lines 444-445 of current manuscript).

The title of section 2 uses the term "multi-drug resistance", which I found confusing, as the real multidrug resistance was discussed in only few cases. The term means resistance to different drugs at the same time, a setting rarely established and almost never tested in literature under discussion. The authors agree that this term may be ambiguous. The text has been modified to replace the term ‘multidrug resistance’ with ‘drug resistance’ (see page 10, line 226 of revised manuscript).

Overall changes to the manuscript format:

Based on reviewer's recommendations, after our first submission, we have inserted additional references (40 additional references, now 134 references total), and incorporated all corrections (with one exception) offered by both reviewers. Additionally, the manuscript has been formatted as requested per the Manuscript Preparation Checklist, and is submitted using the manuscript template. In summary, we believe that our improved and significantly revised manuscript, which describes key progress made in recent CRISPR cancer screens, will be directly relevant to the readers of *Advanced Genetics*. We thank you in advance for your time and careful consideration of our work and look forward to hearing from you.

Yours sincerely,

Dr Marina Pajic

2 nd Peer Review	06 Oct 2022
-------------

Dear Prof. Pajic,

Thank you for submitting your revised Review manuscript entitled "A CRISPR path to solving drug resistance: targeting the diverse cancer landscape and its ecosystem" (Review, No. ggn2.202200014R1) to *Advanced Genetics*. One of the original reviewers has now taken the time to evaluate this submission again. They felt the changes made had improved this paper, but they have some final comments (at the end of this email), which we would invite you to take into consideration in a final revision of this work.

As you revise your manuscript, please also consider the following editorial suggestions:

1. Please ensure that paragraphs are indented or separated by a newline.
2. I would encourage you to consider breaking up some of the longer paragraphs to help improve readability. For example the long paragraph stretching across pages 12 and 13 could perhaps be broken at "For instance ..." and "Moreover", and these sentence could be slightly rewritten to serve as strong topic sentences.
3. Please take this final opportunity to thoroughly review your text for grammar and clarity. Please pay special attention to complex sentences, especially ones with participle phrases, and consider whether shorter or more direct wording may be clearer. I note a few examples of minor issues below:

line 144: "broad range of cancers.[51] and" (remove period)

line 145: "...harbor MSI and associated significantly higher burden of mutations may..." (This reads somewhat awkwardly. Could it be better written as "have MSI and a higher burden of mutations may...")

line 311-312: "One way in which the secondary resistance may appear involves epigenetic alterations

arising on-treatment, resulting in..." (This may be better written as "One way in which secondary resistance can appear during treatment is through epigenetic alterations that result in...")
line 452-454: "Additionally, Ringel et al[131] described a method of sequencing individual sgRNA-transduced organoids, allowing whole-genome screens to be performed without the need for large cell numbers." (This is another example of a sentence that might be clearer without the participle phrase, e.g. "Additionally, Ringel et al[131] described a method for sequencing individual sgRNA-transduced organoids that allows whole-genome screens to be performed without the need for large cell numbers.")

To submit your revision, go to <https://www.editorialmanager.com/advgenet/> and log in as an Author using your username (*****) and password. Your submission can be found under the menu item "Submissions Needing Revision". The changes to your manuscript should be highlighted in a different color in the primary "Revised Manuscript" file.

With your revision, please provide another point-by-point response letter covering both my editorial comments and the final comments from Reviewer 2. As these are all relatively minor issues, I will seek make a rapid decision without further peer-review.

We should receive your revised manuscript by 16 Oct 2022. Once we receive your revised manuscript, we will provide a final decision as soon as possible.

Should you need extra time, do not hesitate to contact the editorial office.

Yours sincerely,

Andrew Hufton

--

Dr Andrew Hufton, Editor
Advanced Genetics
E-mail: AdvGenet@wiley.com
Tel: +49(0)6201-606-362

<http://www.advgenet.com>

P.S. Please help avoid delays by referring to the Manuscript Preparation Checklist (<http://www.advgenet.com/authorguidelines>) and use the appropriate article template when preparing your revised manuscript. Please also follow the instructions to prepare and upload your Production Data materials. These include: the full, non-highlighted text of your manuscript (with all figures with a resolution of at least 300 dpi, schemes, and tables) in editable format - Word DOC/X or LaTeX; a short summary (50-60 words) and an eye-catching color image for the Journal's Table of Contents; and, if applicable, Supporting Information*.

Copyright information should be included in each figure caption as follows (the reference number [REF] should be superscripted):

Reproduced (Adapted) with permission.[REF] Copyright YEAR, Publisher Name.

Or, for figures reproduced or adapted from CC-BY open access publications (insert the name of the specific CC-BY license for XXX; permission should be obtained from the Copyright Holder for CC-BY-NC licenses):

Reproduced (Adapted) under the terms of the XXX license.[REF] Copyright YEAR, Copyright Holder Name(s).

REVIEWER REPORT:

COMMENTS TO AUTHOR:

Reviewer #2: The authors submit a new, significantly improved version of the manuscript entitled: "A CRISPR path to solving drug resistance: targeting the diverse cancer landscape and its ecosystem". Most of my concerns have been satisfactorily addressed and the manuscript reads clearly better. Apart from two minor concerns, I want to first address the problem of manuscript structure. As the initial title suggested, targeting cancer drivers was according to the authors a "path to solving cancer resistance". This could be understood in two ways: targeting drivers of cancer resistance, or targeting drivers of cancer growth (independent of the therapy). The latter usage is common in the field and is the one I based my critique on. There is simply no evidence that targeting drivers in this latter sense will reverse resistance. Accordingly, the authors have changed the title, so that targeting drivers is now removed. Although I find the new title a better fit to the presented text, the relationship of section 2 to the rest of the manuscript is still a little awkward. If the authors both in the title and in the introduction (section 1) concentrate solely on the question of resistance, why do they claim they are interested in general vulnerabilities? The authors: "whether a given gene is a 'new vulnerability' or a 'resistance mechanism' is not really relevant, as from a pragmatic point of view both 'new vulnerabilities' and 'resistance mechanisms' could be targeted to successfully improve disease outcome." I agree with this statement. But then I strongly suggest the title should read "A CRISPR path to finding vulnerabilities and solving drug resistance: targeting the diverse cancer landscape and its ecosystem", and the authors should make the concept of vulnerability in this broader sense (not only resistance vulnerability) apparent in the introduction. The lines 77-80 are a good place to do so.

Minor concerns:

1. the problem with screen nomenclature is now solved in the text, but not in the Figure 1. Terms

"positive" and "negative" screen should be corrected.

2. A reference for prime editing is missing, line 104. The reference 30 is incomplete.

Authors' Response to 2nd Review

11 Oct 2022

Dear Dr Kerstin Brachhold,

Re: Advanced Genetics - New Submission of Revised Manuscript 'A CRISPR path to solving drug resistance: targeting the diverse cancer landscape and its ecosystem' (Review, No. ggn2.202200014)

We would like to thank you for the final comments on our revised manuscript by reviewer 2 and further associated editorial comments. We have incorporated the final changes, and we believe our newly revised manuscript is now in an excellent position to be considered for publication in *Advanced Genetics*.

We list below all of the comments/issues and how we addressed them:

-----Reviewer 2-----

Reviewer 2 "strongly suggests the title should read 'A CRISPR path to finding vulnerabilities and solving drug resistance: targeting the diverse cancer landscape and its ecosystem', and the authors should make the concept of vulnerability in this broader sense (not only resistance vulnerability) apparent in the introduction. The lines 77-80 are a good place to do so".

We have amended the title accordingly, as recommended by Reviewer 2. We introduced the concept of broader vulnerability (line: 79) as suggested by Reviewer 2: 'to **identify critical cancer vulnerabilities, and evaluate mechanisms of** drug resistance or susceptibility'.

Minor concerns:

Problem with screen nomenclature is now solved in the text, but not in the Figure 1. Terms "positive" and "negative" screen should be corrected. The terms 'positive' and 'negative' have been changed to 'enriched' and 'depleted', to remove any potential ambiguity arising from the use of these terms (see figure 1).

A reference for prime editing is missing, line 104. A refence has been added for prime editing (reference 25) (now line 105).

The reference 30 is incomplete. The full author list for this publication comprises a very extensive number of authors, and the citation used by the paper itself simply uses the expression 'The ICGC/TCGA Pan-Cancer analysis of Whole Genomes Consortium' in place of the full author list. In the references section, reference 30 now reads 'The ICGC/TCGA Pan-Cancer Analysis of Whole Genomes Consortium. *Nature* 2020, 578, 82'.

-----Editorial comments-----

1. Please ensure that paragraphs are indented or separated by a newline. This is now corrected in the revised manuscript.

2. I would encourage to consider breaking up some of the longer paragraphs to help improve readability. For example the long paragraph stretching across pages 12 and 13 could perhaps be broken at "For instance ..." and "Moreover", and these sentence could be slightly rewritten to serve as strong topic sentences. Paragraph has been introduced, beginning of Page 13 'For instance,...' – 'a combination of' has been removed to make the sentence more concise and 'punchy', 'new' has been changed to 'novel'. Second paragraph was introduced for the Clements et al study and the sentence was reworked by moving 'additional novel genetic determinants of PARP-i resistance' forward and introducing linkers to strengthen sentence meaning.

In the associated section, Line 311: 'PARP-i therapy may thus have important implications not only...' has been changed to 'PARP-i therapy may thus **play** important **roles** not only...' to improve sentence readability and remove duplication of these terms further down in the sentence.

Line 201: New paragraph has also been introduced and sentence 'Other CSC specific components have also been identified' has now been rewritten. '**Other key CSC specific components have also been comprehensively characterised through integrated genomic approaches**'. Paragraph was further rewritten to strengthen the critical analysis of the work. Specifically sentence: 'Historical analysis revealed predictive power for cancer aggression and metastasis, and when inhibited, cancer growth was reduced, and survival rates extended.' was removed and re-worked to read as follows: '**Genetic or pharmacological block of ROR γ dramatically reduced pancreatic cancer growth in both patient-derived and genetically engineered in vivo settings. Its role in promoting the aggressive nature of pancreatic cancer was further characterised in retrospective clinical samples**'

Line 335: New paragraph has been introduced and an associated topic sentence. '**More recently, global alterations in the chromatin landscape of ER-positive breast cancer have been characterised as a major driver of clinical resistance to ER-targeting therapy**'. Second sentence was also re-worked to strengthen the meaning. Third sentence 'Mechanistically, loss of ARID1A...' 'a member of the SWI/SNF chromatin remodeling family' was moved and introduced in Sentence 2.

Line 405: New paragraph was introduced.

3. Please take this final opportunity thoroughly review your text for grammar and clarity. Please pay special attention to complex sentences, especially ones with participle phrases, and consider whether shorter or my direct wording may be clearer. I note below a few examples of minor issues below:

Line 103: comma error before reference 22 has been changed to full stop.

Line 144: "broad range of cancers.[51] and" (remove period) This has been removed (now line 146).

Line 145: "...harbor MSI and associated significantly higher burden of mutations may..." (This reads somewhat awkwardly. Could it be better written as "have MSI and a higher burden of mutations may..."). Agreed, this has been changed to '**a higher mutational burden**' (now line 147).

Line 187: 'not affected' has been changed to '**unaffected**'

Line 195: 'in the negative regulation' has been changed to '**as negative regulators**' to improve the sentence structure.

Line 265: CRISPR 'screening' has been changed to CRISPR '**analyses**' as screening is repeated again on line 266 in the same sentence.

Line 277: 'notorious' has been corrected to '**notoriously**'

Line 278: 'Aiming' and 'with a specific interest' have been removed to simplify and strengthen the sentence.

Line 311: "One way in which the secondary resistance may appear involves epigenetic alterations arising on-treatment, resulting in..." (This may be better written as "One way in which secondary resistance can appear during treatment is through epigenetic alterations that result in...") This has been corrected as recommended (now line 316).

Line 374: error comma at the end of the sentence was corrected to full-stop.

Line 404: Sentence was made more concise by altering 'causing the tumors to shrink dramatically' to '**leading to dramatic tumor shrinkage**'

Line 452-454: "Additionally, Ringel et al[131] described a method of sequencing individual sgRNA-transduced organoids, allowing whole-genome screens to be performed without the need for large cell numbers." (This is another example of a sentence that might be clearer without the participle phrase, e.g. "Additionally, Ringel et al[131] described a method for sequencing individual sgRNA-transduced organoids that allows whole-genome screens to be performed without the need for large cell numbers."). This has been written as recommended (now line 460).

Based on reviewer's and editorial recommendations, we have made all the changes (these are highlighted in **purple** throughout the revised document). We thank you in advance for your time and careful consideration of our work and look forward to hearing from you.

Yours sincerely,

Associate Professor Marina Pajic

Final Decision	14 Oct 2022
----------------	-------------

Dear Prof Pajic,

Thank you for submitting your revised manuscript entitled "A CRISPR path to finding vulnerabilities and solving drug resistance: targeting the diverse cancer landscape and its ecosystem" (Review, No. ggn2.202200014R2) to Advanced Genetics.

I'm pleased to inform you that your manuscript has been accepted for publication without change.

We will copyedit the accepted version of your manuscript and if we require any further information at this stage we will contact you. After copyediting we will let you know when you can expect to receive the proofs. Instructions for returning your proof corrections will be provided when the proofs are sent to

you.

All articles published in Advanced Genetics are fully open access: immediately and freely available to read, download and share. Advanced Genetics charges a publication fee to cover publication costs. The corresponding author for this manuscript should have already received a quote with the article publication fee, and will soon receive an e-mail invitation to register with or log in to Wiley Author Services (<https://authorservices.wiley.com>). After logging into Wiley Author Services, the publication fee can be paid by credit card, or an invoice or pro forma can be requested. Payment of the publication charge must be received before the article will be published online.

Thank you for choosing Advanced Genetics for publishing your work. I hope you will consider us for the publication of your future manuscripts.

Yours sincerely,

Andrew Hufton

P.S.: If you believe your images might be appropriate for use on the cover of Advanced Genetics, and you would like your paper to be considered for the cover, please e-mail us your layout suggestions with a short description. For details on cover image preparation, please see the cover gallery on <http://www.advgenet.com>.

--

Dr Andrew Hufton, Editor
Advanced Genetics
E-mail: AdvGenet@wiley.com
Tel: +49(0)6201-606-362

<http://www.advgenet.com>